# Impact of a COVID-19 Outbreak in an Elderly Care Home after Primary Vaccination

**DOI:** 10.3390/vaccines11081382

**Published:** 2023-08-19

**Authors:** Elba Mauriz, José P. Fernández-Vázquez, Cristina Díez-Flecha, Sofía Reguero-Celada, Tania Fernández-Villa, Ana Fernández-Somoano, Joan A. Caylà, Jesús A. Lozano-García, Ana M. Vázquez-Casares, Vicente Martín-Sánchez

**Affiliations:** 1ALINS, Food Nutrition and Safety Group, ICTAL Universidad de León, 24007 Leon, Spain; 2Department of Nursing and Physiotherapy, Campus de Vegazana, Universidad de León, s/n, 24071 Leon, Spain; ana.vazquez@unileon.es; 3Health Center Valencia de Don Juan, Primary Health Care Management, 24200 Leon, Spain; jpfvfg@gmail.com; 4Primary Health Care Management, 24008 Leon, Spain; cridifle@saludcastillayleon.es; 5Health Center San Andrés de Rabanedo, Primary Health Care Management SACYL, 24191 Leon, Spain; sofiareguero@gmail.com; 6Group of Investigation in Interactions Gene-Environment and Health (GIIGAS), Institute of Biomedicine (IBIOMED), Universidad de León, 24071 Leon, Spain; tferv@unileon.es (T.F.-V.); vicente.martin@unileon.es (V.M.-S.); 7Spanish Consortium for Research on Epidemiology and Public Health (CIBERESP), 28029 Madrid, Spain; fernandezsana@uniovi.es; 8IUOPA–Department of Medicine, University of Oviedo, Julián Clavería Street s/n, 33006 Oviedo, Spain; 9Instituto de Investigación Sanitaria del Principado de Asturias (ISPA), Roma Avenue s/n, 33001 Oviedo, Spain; 10Tuberculosis Research Unit Foundation of Barcelona, 08008 Barcelona, Spain; joan.cayla@uitb.cat; 11MC MUTUAL Group, 24002 Barcelona, Spain; spress68@hotmail.com

**Keywords:** initial vaccination, elderly care home, pre-existing medical conditions, nursing home, vaccination coverage

## Abstract

Elderly care home residents are particularly vulnerable to COVID-19 due to immune-senescence, pre-existing medical conditions, and the risk of transmission from staff and visitors. This study aimed to describe the outcomes of a COVID-19 outbreak in a long-term care facility for elderly persons following the initial vaccination. A single-center, retrospective, observational design was used to analyze the variables associated with hospitalization and death rate by logistic regression. Adjusted odds ratios (aOR) and their 95% confidence intervals (CI) were calculated. Sixty-eight residents received the first dose of the COVID-19 vaccine. Despite being negative six days after vaccination, the performance of a second test 4 days later revealed 51 positives (75.0%) among residents and 18 among workers (56.3%). A total of 65 of the 68 residents (95.58%) had positive results with symptoms, whereas 34.9% required hospitalization, and 25.8% died. The best-fitting model to explain the distribution of cases reflects three points at the time of infection.. The time from vaccination to symptom onset explains the hospitalization and mortality rates since a day elapsed halves the risk of hospitalization (aOR = 0.57; CI = 0.38−0.75) and the risk of death by a quarter (aOR = 0.74; CI = 0.63−0.88). Nursing homes present an elevated risk of transmission and severity of SARS-CoV-2 infection. Although vaccination reduces the risk of hospitalization and death, extreme prevention and control measures are essential in these institutions despite the high vaccination coverage.

## 1. Introduction

Elderly long-term care facilities (LTCF) are particularly conducive to SARS-CoV-2 infection transmission owing to the distinctive characteristics derived from communal living and the age and comorbidities of the residents, thus resulting in high case-fatality rates (CFR) [1]. Even before the coronavirus disease 2019 (COVID-19) pandemic, outbreaks of respiratory infections severely affected LTCFs. 

The surveillance of COVID-19 in LTCFs in the EU/EEA and the US has reported outbreaks in more than half of these centers [2,3,4], with reported cases from 22% to 47% in residents and between 24% and 45% in staff during the period of the COVID-19 pandemic previous to the approval of the vaccine. The CFR ranged from 27% to 36.9% in residents [5]. In Spain, the weekly report on COVID-19 in residential centers ceased to be published in February 2023. This report displayed that the rate per 10,000 residents confirmed with COVID-19 by diagnostic tests over the total number of residents ranged from 190.92 in January 2022 to 18.6 in January 2023, while the CFR was 7.65% and ranged between 20.39% in 2020 and 2.59% in 2023 [6].

These high CFRs drove both the European and United States Centers for Disease Control and Prevention (ECDC and CDC) to recommend, in December 2020, the priority vaccination of LTFC residents and healthcare workers during the early pandemic [7]. Following the European Union vaccine strategy to ensure rapid and equitable access to vaccines by all member States, the COVID-19 Vaccination Technical Working Group and the Vaccine Committee coordinated by the Inter-territorial Council of the National Health System (CISNS) adopted, in Spain, a set of measures to prioritize the vaccination of population groups according to an established ethical framework and risk criteria. Based on the risk of severe morbidity and mortality, exposure, socioeconomic impact, and transmission, the group prioritization established that the first stage of vaccination might include residents and health and social care personnel in care homes for the elderly and the disabled [8]. The efficacy of the currently available mRNA vaccines against SARS-CoV-2 to prevent infections after two to three weeks of the first dose can achieve an efficacy of 57% (95% confidence interval [CI], 50–63%) overall, 44% (CI: 19–64%) among individuals ≥70 years of age, and 62% (CI: 43–77%) among those presenting at least three co-morbid conditions [9]. Therefore, single-dose vaccination can reduce transmission by 40–50% [10] and reach 90% effectiveness in preventing severe and fatal cases. The European Medicines Agency authorized the use of the mRNA vaccines according to their vaccine efficacy information [11]. Since the staff and visitors are the primary transmission sources [12,13], outbreaks continue to emerge in LFTC even when most of the residents (97.3%) and staff are fully vaccinated [14,15,16,17]. For all these reasons, the last guidelines enforce the need to maintain and maximize all preventive and control measures in LTFC to reduce the risk of infection after starting the vaccination process and the danger that may arise from the false sense of security after the administration of the first dose. For instance, to interrupt transmission and prevent dissemination, the European Geriatric Medicine Society and CDC recommend weekly screening programs until they reached a 70% vaccination rate [18] and 3–7 days of testing when a case of COVID-19 is detected [7,12]. Accordingly, new preventive strategies focus on the long-term effects of vaccination in LTFC residents and other subgroups at higher risk of COVID-19 [17]. Therefore, this study aims to evaluate the outcomes in an elderly care facility where, after the first dose of the vaccine, an outbreak with high infection fatality rates occurred a few days later.

## 2. Methods

### 2.1. Study Design and Setting Description

We conducted a retrospective observational study on an elderly care facility in León, Spain. We investigated the epidemiological and clinical characteristics of a COVID-19 outbreak between 30 December 2020 and 15 February 2021 (14 days after the last case).

The facility consists of one building, with a total capacity of 74 residents living in single or double rooms. The residence had a surface of 3785 m^2^. The occupancy percentage was 92% (68 of 74) during the outbreak. The number of workers at that moment was 35. The staff composition according to the category was 25 health workers (21 geriatricians, one full-time nurse and occupational therapist, one medical doctor/general practitioner), one physiotherapist, and one social worker); the director-manager; one part-time hairdresser; three kitchen staff; three cleaners; and one maintenance worker. The care facility followed the recommendations on vaccination and control measures contained in the protocol for the epidemiological surveillance of residential centers in the EU/EEA countries coordinated by the European Centre for Disease Prevention and Control (ECDC) [2], as well as the recommendations to nursing homes and social-health centers for COVID-19 from the Spanish government [19]. These measures included not only actions to be taken in case of contacts and cases of COVID-19 but also general measures for the protection of workers’ health including hand hygiene, the use of personal protective equipment (e.g., gloves, masks, eyewear), the cleaning and disinfection of surfaces and spaces, waste management, crockery and linen, and the identification of contacts of cases under investigation. The strict anti-COVID protocol in the facility also included the exclusive utilization of the lounge for dependent residents and a second lounge converted into a dining room for the dependents. Meal service occurred in two shifts, with only residents sharing a table in the same room. At the beginning of 2021, each resident was allowed to receive two visits, which took place in a gallery equipped outside the walls of the Residence.

### 2.2. Data Collection: Case Definition, Testing Strategy and Follow-Up Residents

The collection of clinical information occurred as part of routine (public health) surveillance for COVID-19. A chronological account of the events related to this outbreak, including the epidemic curve, was made.

In Spain, confirmed SARS-CoV-2 infections reported to the regional Public Health Service necessitate a protected online form containing data about demographics, the date of symptoms onset and test, type of ward, hospital admission due to COVID-19, or death within 30 days of testing positive, previous COVID-19 episodes, and vaccination status (vaccine type, number of doses, dates of administration).

Following the regional protocol, a confirmed case of a COVID-19 infection was a person with laboratory confirmation of the virus causing COVID-19 infection via a molecular test (PCR or rapid antigen test), irrespective of clinical signs and symptoms [20,21]. After a confirmed case, testing all residents was mandatory every five days until finding no more positives. 

### 2.3. Measures and Data Analysis

The detection of SARS-CoV-2 in upper respiratory samples using nucleic acid amplifications tests (NAATs) involved either RT-PCR targeting E, RdRP/S, and N regions (AllplexTM SARS-CoV-2, Seegene Inc, Seoul, Korea) or transcription-mediated amplification (TMA), targeting the N region (Procleix SARS-CoV-2, Grifols, Sant Cugat del Vallès, Barcelona, Spain).

A chronological account of the events related to this outbreak, including the epidemic curve, was made. The estimation of case distribution was made according to the incubation periods reported by McAloon et al. [22], calculating the proportions of hospitalization and death with their 95% CI and their distribution by age, sex, estimated date of infection, time from vaccination to onset of symptoms, and the cycle threshold of RT-PCR. Using unconditional logistic regression, we estimated the risks of hospitalization and death adjusted for sex, age, time from vaccination to symptom onset, and PCR threshold cycles.

### 2.4. Ethical Consideration

The research protocol of this study was approved by the Ethical Committee for Research with Medicines of the León and Bierzo Health Areas (CEIm) (P.I. 20122). Informed consent was not required. The information safety commission provided approval. The study followed the principles of the Declaration of Helsinki.

## 3. Results

On 30 December 2020, the vaccination of residents and workers with the first dose of the BNT162b2 mRNA vaccine (Comirnaty©) began.

After vaccination, many residents presented symptoms such as a cough, mucus, and prostration, generally attributed to the vaccine’s adverse effects. Given this situation, on 5 January 2021, an antigen test was carried out on all the residents, with negative results. On 8 January 2021, 68 residents vaccinated on 30 December 2020 were at the center. That same day, one of the residents presented symptoms compatible with COVID-19 and tested positive in a rapid antigen detection test. On 9 January 2021, three more residents presented positive results in the antigen detection tests. On this day, samples for RT-PCR taken from the residents of the center and the workers resulted in 51 of the 68 residents (75.0%) being found positive, while 18 of the 32 workers (56.3%) tested positive. At that time, the residence was sectorized, leaving one floor for positive cases and another for negative ones.

The 17 negative cases among the residents were re-sampled after seven days, and 14 tested positive. Altogether, 65 of the 68 residents (95.58%) were infected with SARS-CoV-2. All 65 residents with PCR+ had symptoms. Figure 1 shows the distribution of cases by the date of symptom onset. The median number of days from vaccination to symptom onset was 23 (P25–P75: 18–25 days; mean: 21.1 days; 95% CI: 19.8–22.3). Regarding the incubation period of SARS-CoV-2 infection, the model that best fits the distribution of cases is that of three different infection times on 6, 11, and 18 January 2021.

Table 1 shows the overall incubation periods and those of the three outbreaks, with a median of 5 days and a mean of 4.9 days.

Figure 2 and Figure 3 show the epidemic curve of symptomatic cases, hospital admissions and deaths, and the case-by-case evolution.

Of the 65 residents with PCR+ presenting symptoms, 23 (34.9%; CI: 23.5–47.6%) required hospital admission and 17 (25.8%; CI: 15.8–38.0%) died. While 100% (8/8) of the residents involved in the first peak required hospital admission and 87.5% (7/8) died, the percentages of hospitalized and deceased were reduced in the second [84.6% (11/13) and 46.2% (6/13)] and third peak [8.9% (4/45) both hospitalized and deceased].

Table 2 shows how the cycle time (Ct) results varied depending on when the PCR was positive and the assignment of the residents to each of the outbreak waves. Thus, the lowest cycles corresponded to the residents of the first wave (on 9 January 2021), where all were positive in the first PCR, and the highest to the residents who were positive in the third wave. Similarly, Ct values were lower in residents who tested positive in the second PCR than in those who tested positive in the first PCR.

Table 3 shows the risks of hospitalization and death according to the chosen independent variables. Although men and older patients had a higher risk of hospitalization and death, they did not reach statistical significance in the multivariate analysis. Similarly, those with higher cycles had a lower risk of hospitalization and death but did not reach statistical significance in the multivariate analysis. Only the time elapsed from vaccination to the onset of symptoms remains a factor associated with hospitalization and death. Thus, the risk of hospitalization is reduced by almost half and the risk of death by more than a quarter (Table 3) for each day elapsed from vaccination to the onset of symptoms.

## 4. Discussion

This study described the rapid transmission of COVID-19 in elderly persons residing in a nursing home despite initial vaccination. Our data suggest that a single dose of the BNT162b2 vaccine did not protect older patients against COVID-19 infection well. The severity of the outbreak among the residents resulted in a high overall SARS-CoV-2 infection attack rate, hospitalization, and case fatality figures. We observed an attack rate between 75% (first peak) and close to 100% (third peak), even superior to the data reported in other COVID-19 outbreaks of long-term care facilities, wherein the average attack rates were in the 50–70% range.

In a systematic review, Hashan et al. found an average attack rate of 45% [95% CI 32–58%] after initial vaccination, lower than that produced in the center under study. On the other hand, high attack rates have also been described, even after vaccination [23]. For example, Van Ewij et al. investigated the vaccine effectiveness under a high COVID-19 vaccination coverage among the residents (92%) and strict hygiene preventive measures [1]. This work explains it as a result of the high force of infection over a large outbreak, deficient hygiene measures, or emerging SARS-CoV-2 variants associated with lower vaccine effectiveness [1,24]. In this sense, the high attack rate observed among workers was consistent with persistent exposure and the less restrictive application of preventive measures just after vaccination began. Thus, from the analysis of possible incubation periods, this was not a single exposure to a source of infection, most likely at least three exposures to infection sources or continuous transmission for at least 14 days [22]. The Christmas holidays and the return from weekends sustained the hypothesis of multiple exposures. Furthermore, it coincides with high community transmission (Basic reproduction number (R0): 9 January 2021; R0: 1.89 León), and the entry of the B.1.1.17 strain and its increased transmissibility.

The high prevalence of infection in residents and workers makes it difficult to know whether transmission occurred from workers to residents or vice versa. The staff might contribute to the rapid circulation of the virus due to non-ordinary circumstances, such as the family gatherings surrounding the time of infection. Additionally, the maintenance at work of staff with early minor symptoms could result in delayed identification. Recent vaccination and the possible adverse effects, and the atypical clinical presentation in those vaccinated, could also have affected the delay in taking measures and the late identification of the index case (10.5 days after vaccination) [25,26]. Hashan et al. reported that the disease transmission was almost half, and was half once the index case was defined. Other outbreaks in nursing care homes involving infected staff have shown rapid COVID-19 spread with higher infection [OR = 4.2 (CI: 2.6 to 6.8)], hospitalization [OR = 2.8 (CI: 1.7 to 4.7)] and death [OR = 2.2 (CI: 1.3 to 3.7)] rates [12]. This fact revealed the importance of maintaining strict hygiene controls regardless of initial vaccination since engineered and procedure controls and the appropriate use of personal protective equipment (PPE) outweigh the role of vaccination for the infection prevention and control of COVID-19 in elderly nursing homes.

These findings also suggest the shortening of incubation periods due to the residents’ precarious immune and general state and the high viral load circulating in the facility.

Age and associated immunosenescence, and a high prevalence of comorbidities largely explain the need for hospitalization and the high number of fatality cases in elderly care settings [27]. In our center, the hospitalization (34.9%) and CFR (25.8%) figures were lower than those reported by other authors, with hospitalization rates of up to 60% [27,28] and case fatality rates of up to one-third of those infected in unvaccinated residents, although very similar to that found in a previous meta-analysis (37% [CI: 35–39%] and 23% [CI: 18–28%]) [23] and another monitored outbreak, with lower hospitalization rates (28%) and slightly higher fatality rates (29%) [25].

Despite the outbreak severity, since all residents received a vaccine dose, the different times of infection could partly attenuate the appearance of symptoms and the incidence and hospitalization rates. A single dose of the vaccine does not reduce either the probability of infection or the occurrence of severe cases or deaths, especially when the time between the administration of the first dose and confirmed infection was less than 10 days. For instance, an analysis of 2239 clusters in Belgian nursing homes revealed that hospitalization rates and CFR decreased drastically after the first dose of the vaccine [21]. However, other studies reporting SARS-CoV-2 infections shortly after BNT162b2 vaccination showed lower incidence rates in nursing home residents who received only one dose compared to those receiving both doses [29,30]. The relative risk of COVID-19 infection was approximately one-third of the risk in fully vaccinated residents [30].

Although clinical trials of the BNT162b2 vaccine have reported an efficacy of 52% after administration of the first dose [31,32], these efficacies appeared between 13 to 24 days after immunization of vaccination [32]. Additionally, other studies did not find a significant difference in the incidence of infection between the fifth and the twelfth day after receiving the first dose of the vaccine between the vaccinated and unvaccinated groups [33].

Our outcomes also coincide with a previous study that reported a higher incidence within 14 days after the 1st dose of the COVID-19 vaccine and is consistent with the emergence of cases in a short period [7,30]. This fact was also compatible with the reduction of vaccine effectiveness associated with the appearance of variant strains and the slackening of preventive measures after primary vaccination [34,35]. In January 2021, the B.1.351 variant was present concomitantly with the VOC B.1.1.7 (Alpha) in Spain, before becoming predominant in April 2021 [18,36]. Despite the immune cellular response conferring protection after BNT162-b2, the humoral response can reduce the neutralization activity for the B.1.351 variant [37]. However, since the BNT162-2 vaccine is effective against both B.1.1.7 and B.1.351 variants [38], the weaker response after vaccination could be explained by the resistance to serum neutralization of the replacing variant and the susceptibility of a population with multiple co-morbidities [7].

Although the age of the residents might contribute to a declining immunity resulting in reduced antibody response, the BNT162b2 vaccine was not entirely ineffective among residents as the occurrence of hospitalizations and deaths diminished as the days from vaccine to symptoms increased [28]. Similarly, the distribution of SARS-CoV-2 PCR cycle threshold values according to the time of infection provided higher values for the second and third transmission peaks. These results confirmed the effectiveness of the BNT162b2 vaccine in older persons reported by several studies [39] and the likelihood of lower nasopharyngeal viral load after the first vaccine dose compared with a lack of vaccination [7,21]. Our data suggest that receiving the first vaccine dose lessened the disease severity and the risk of dying from COVID-19 [28]. Indeed, according to our findings, the first dose of vaccine reduced the severity of the outbreak, the probability of hospitalization by about 50% and the lethality probability by 25%, for each day elapsed from the administration of the first dose to the appearance of the first symptoms, thus reducing the lethality from 100% and hospitalization from 87.5% in those who presented symptoms earlier, to less than 8.75% in the residents with symptoms later. This finding indicates that vaccination may reduce hospitalization and intensive-care-unit admissions, as reported by other studies [40]. However, a single dose of the vaccine did not result in a drastic decline in CFR. Therefore, primary vaccination seemed to reduce the risk of developing severe disease but was ineffective in avoiding the infection or blocking its transmission.

This study has some limitations. First, although the outbreak impacted residents and staff, data related to the follow-up of the disease of the latter were scarce. Second, the sample size was too small to adjust for potential confounders when estimating vaccine effectiveness and could affect the significance of results. For instance, exposure among wards was not measured, and the observation period involved one month, from the 1st COVID case to 2 weeks after the last COVID case occurred in this facility. Third, although the index case was the first resident with a positive COVID-19 test result, a delay in diagnosis could occur due to residents being asymptomatic or having mild symptoms. Finally, the alpha variant with enhanced transmissibility was the dominant strain of the current COVID-19 pandemic. However, this outbreak coincided with the emerging delta variant, thus interfering with the data stratification.

## 5. Conclusions

This study highlights that COVID-19 outbreaks can still occur despite primary vaccination in elderly facilities. The extremely high attack rates suggest that the first dose of the BNT162-b2 vaccine helped reduce the disease severity, hospitalization, and risk of dying from COVID-19. However, the partial vaccination of residents did not seem to avoid infection or block transmission. Several factors appeared to induce a large outbreak and a high prevalence among residents and staff. Our findings indicate the strong association between immunosenescence, suboptimal adherence to nonpharmaceutical interventions, and decreased vaccination efficacy due to the emergence of SARS-CoV-2 variants.

Therefore, the spread of SARS-CoV-2 infection requires all preventive measures once the vaccination process has begun, devoting specific resources to combating the false sense of security often perceived associated with elderly persons. Further research is needed to improve the effectiveness of SARS-CoV-2 vaccines and limit the transmission of infectious diseases in elderly persons living in long-term facilities.

## Figures and Tables

**Figure 1 vaccines-11-01382-f001:**
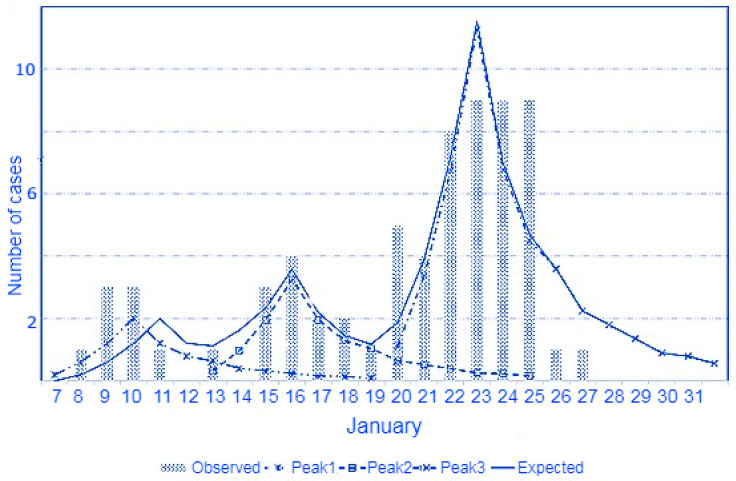
Distribution of COVID-19 cases in residents according to date of symptom onset and simulation of times of infection based on the incubation periods reported by McAloon et al. [22].

**Figure 2 vaccines-11-01382-f002:**
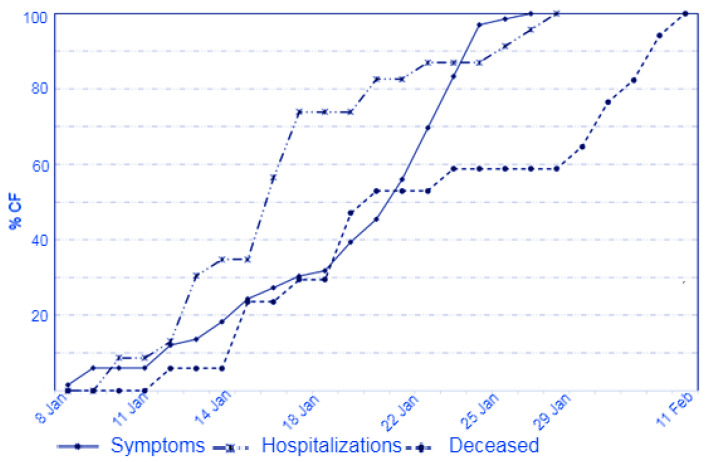
The epidemic curve of symptom onset, hospitalizations, and deaths (%CF = percentage of cumulative frequency).

**Figure 3 vaccines-11-01382-f003:**
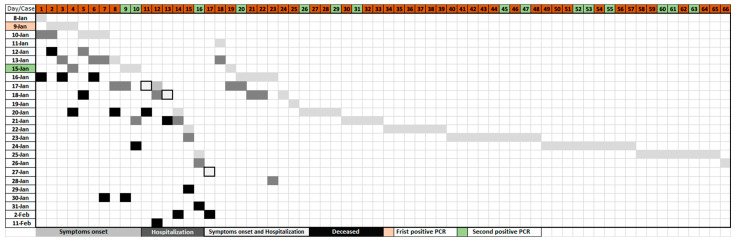
Evolution of cases in residents according to the symptoms onset, hospitalization, and death.

**Table 1 vaccines-11-01382-t001:** Incubation periods and days elapsed from vaccination to onset of COVID-19 symptoms in residents according to the various peaks.

	*n*	Incubation Time		Days from Vaccination to Symptoms	
		Me (P25–P75)	Mean (SD)	Me (P25–P75)	Mean (SD)
Peak 1	8	3.5 (3–4)	3.5 (0.9)	10.5 (10–11)	10.5 (0.93)
Peak 2	13	5.0 (4–6)	5.2 (1.6)	17.0 (16–18)	17.2 (1.6)
Peak 3	45	5.0 (4–6)	5.1 (1.8)	24.0 (23–25)	24.1 (1.8)
Pooled	66	5.0 (4–6)	4.9 (1.7)	23.0 (18–25)	21.1 (5.1)

Me: median; SD: standard deviation.

**Table 2 vaccines-11-01382-t002:** Distribution of threshold cycle time (Ct) values according to time of PCR+.

	*n*	First PCR+			Second PCR+			Pooled	
		*n*	M (P25–P75)	µ (SD)	*n*	M (P25–P75)	µ (SD)	M (P25–P75)	µ (SD)
Peak 1	8	8	22.5 (20–24)	23.1 (5.0)	0			22.5 (20–24)	23.1 (5.0)
Peak 2	13	10	27.0 (21–39)	29.2 (8.7)	3	20.0 (20.0–36.0)	25.3 (9.2)	25.0 (21–38)	28.3 (8.6)
Peak 3	45	33	31.0 (23–39)	30.3 (8.0)	12	24.5 (22.5–27.5)	26.0 (6.5)	27.0 (23–39)	29.2 (7.8)
Pooled	66	51	26.0 (22–39)	29.0 (8.0)	15	24.0 (20.0–28.0)	25.9 (6.7)	25.0 (22–38)	28.3 (7.8)

M: median; µ: media; SD: standard deviation.

**Table 3 vaccines-11-01382-t003:** Distribution of hospitalized and deceased residents according to the selected variables.

	Hospitalized		Odds Ratio			
	*n*	%	OR	CI 95% Range	aOR	CI 95% Range
Sex						
Female (50)	16	32.0	1		1	
Male (16)	7	43.8	1.65	0.52–5.23	4.14	0.46–36.96
	No	Yes				
Age Mean (SD)	87.3 (5.8)	90.9 (5.0)	1.14	1.02–1.28	1.17	0.96–1.44
Days to Symptoms Mean (SD) *	23.8 (1.9)	16.0 (5.2)	0.57	0.44–0.74	0.54	0.38–0.75
Ct-RT-PCR Mean (SD)	29.8 (7.8)	25.4 (7.1)	0.92	0.86–0.99	0.88	0.77–1.00
	Deceased		Odds Ratio			
Variables	*n*	%	OR	CI 95% range	aOR	CI 95% range
Sex						
Female (50)	10	20.0	1		1	
Male (16)	7	43.8	3.11	0.93–10.4	4.54	0.75–27.3
	No	Yes				
Age Mean (SD)	88.1 (6.0)	90.9 (5.0)	1.06	0.95–1.17	1.04	0.91–1.18
Days to Symptoms Mean (SD) *	22.9 (3.2)	15.9 (6.0)	0.74	0.63–0.86	0.74	0.63–0.88
Ct-RT-PCR Mean (SD)	29.3 (7.8)	25.4 (7.3)	0.93	0.86–1.01	0.94	0.85–1.04

* Days from vaccination to symptoms; aOR: adjusted OR.

## Data Availability

Data will be available upon reasonable request to the corresponding author.

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
