# Peer review of "Impact of a COVID-19 Outbreak in an Elderly Care Home after Primary Vaccination"

_vaccines, 2023, doi:10.3390/vaccines11081382_

Round 1

Reviewer 1 Report

Dear Author,

Greetings!

1 Elderly care home residents are particularly vulnerable to COVID-19 due to immunose-28 nescence, pre-existing medical conditions, and the risk of transmission from staff and visitors. This 29 study aimed to describe an outbreak of COVID-19 in a nursing home after the initial vaccination(Please revise this)

2the performance of a second test four days 34 later revealed 51 positives (75.0%) among residents and 18 among workers (56.3%). One week later, 35 14 residents of the 17 negatives resulted positive. In total, 95.58% were affected, 34.9% required 36 hospitalization, and 25.8% died. The best-fitting model to the distribution of cases reflects three 37 points at the time of infection.(It is not clear please modify and highlight the significance of specific immunity) 

3Surveillance of COVID-19 in LTCF in the EU/EEA and the US have reported out- 53

breaks in more than half of these centers [2–4], with incidences from 22% to 47% in resi- 54

dents and between 24% and 45% in staff. The CFR ranged from 27% to 36.9% in residents 55

[5]. In Spain, the rate per 10,000 residents confirmed with COVID-19 by diagnostic tests 56

over the total number of residents has ranged from 190.92 in January 2022 to 18.6 in Janu- 57

ary 2023, while the CFR was 7.65 % and ranged between 20.39% in 2020 and 2.59% in (Please add updated information)

4 PLease modify Study design and setting description 82

We conducted a retrospective observational study on an elderly care facility in León, 83

Spain. We investigated the epidemiological and clinical characteristics of the cohort of a 84

COVID-19 outbreak between 30 December 2020 and 15 February 2021 (14 days after the 85

last case). 86

The facility consists of one building, with a total capacity of 74 residents living in 87

single or double rooms. The useful square meters of the residence were 3785. The occu- 88

pancy percentage was 90% (70 of 74) during the outbreak. The staff included 21 geriatri- 89

cians, one full-time nurse and occupational therapist, one medical doctor (general practi- 90tioner), one physiotherapist, one social worker and one part-time hairdresser, the Direc- 91tor-Manager, three kitchen staff, three cleaners, and one maintenance worker. The strict 92anti-COVID protocol in the facility included the exclusive lounge utilization for depend- 93ent residents and a second lounge converted into a dining room for the dependents. (ITS NOT CLEAR)

7A chronological account of the events related to this outbreak, including the epidemic 117 curve, was made. The estimation of cases distribution was according to the incubation 118 periods reported by McAloon et al. [18], calculating the proportions of hospitalization and 119 death with their 95% CI and their distribution by age, sex, estimated date of infection, time 120 from vaccination to onset of symptoms, and Cycle threshold of RT-PCR. Using uncondi-121 tional logistic regression, we estimated the risks of hospitalization and death adjusted for 122 sex, age, time from vaccination to symptom onset, and PCR threshold cycles. 123

2.4. Ethical consideration 124

The clinical research ethics committee of the Hospital of León approved the study 125 protocol. Informed consent was not required. The information safety commission pro-126 vided approval. The study followed the principles of the Declaration of Helsinki.(A chronological account of the events related to this outbreak, including the epidemic 117 curve, was made. The estimation of cases distribution was according to the incubation 118 periods reported by McAloon et al. [18], calculating the proportions of hospitalization and 119 death with their 95% CI and their distribution by age, sex, estimated date of infection, time 120 from vaccination to onset of symptoms, and Cycle threshold of RT-PCR. Using uncondi-121 tional logistic regression, we estimated the risks of hospitalization and death adjusted for 122 sex, age, time from vaccination to symptom onset, and PCR threshold cycles. 123

2.4. Ethical consideration 124The clinical research ethics committee of the Hospital of León approved the study 125 protocol. Informed consent was not required. The information safety commission pro-126 vided approval. The study followed the principles of the Declaration of Helsinki.(PLEASE PROVIDEethical clearance AUTHORISED  CERTIFICATE AND UPDATE WITH NEW INFORMATION)

9On the 9th, three more residents presented positive results in 136 the antigen detection tests. On this day, samples for RT-PCR taken from the residents of 137 the center and the workers resulted in 51 of the 68 residents (75.0%) being found positive 138 while 18 of the 32 workers (56.3%) tested positive. At that time, the residence was sector-139 ized, leaving one floor for positive cases and another for negative ones. 140

The 17 negative cases among the residents were re-sampled after seven days, and 14 141 tested positive. Altogether 65of the 68 residents (95.58%) were infected with SARS-CoV-142 2. All 65residents with PCR+ had symptoms. Figure 1 shows the distribution of cases by 143 date of symptom onset.(PLEASE MODIFY )

FIGURE 1 AND FIGURE 2 PLEASE CHANGE COMPLETELY (please add hours and logic of taking 7,11 please explain)

11,Data availability statement is missing 

12 Pleaee arrange the references clearly as per journal format and check with mendeley 0r zotero 

Best 

only few corrections in introduction

Author Response

We thank the reviewers for their constructive and valuable suggestions. Their careful reading has contributed to improving the quality of this article. We have comprehensively revised the manuscript to address all their requests. Following the editor’s instructions, we have included a point-by-point response to the reviewers’ comments, and the changes made have been highlighted accordingly in the revised version of the manuscript.

To help review amendments done to the manuscript, below we reproduce the reviewer’s comments (black color and italics font), and immediately after, the reply is provided in red color and portions of the revised text in the paper between quotation marks (red color and italics).

Reviewer 1

1 Elderly care home residents are particularly vulnerable to COVID-19 due to immunose-28 nescence, pre-existing medical conditions, and the risk of transmission from staff and visitors. This 29 study aimed to describe an outbreak of COVID-19 in a nursing home after the initial vaccination(Please revise this)

We thank the reviewer for the comment. Following the recommendation, this statement has been revised to:

Elderly care home residents are particularly vulnerable to COVID-19 due to immunosenescence, pre-existing medical conditions, and the risk of transmission from staff and visitors. This study aimed to describe the outcomes after a COVID-19 outbreak in a long-term care facility for elderly persons following the initial vaccination”.

2 the performance of a second test four days 34 later revealed 51 positives (75.0%) among residents and 18 among workers (56.3%). One week later, 35 14 residents of the 17 negatives resulted positive. In total, 95.58% were affected, 34.9% required 36 hospitalization, and 25.8% died. The best-fitting model to the distribution of cases reflects three 37 points at the time of infection.(It is not clear please modify and highlight the significance of specific immunity)

We appreciate the suggestion. To clarify this aspect, the revised statement is as follows:

Despite being negative six days after vaccination, the performance of a second test revealed 51 positives (75.0%) among residents and 18 among workers (56.3%). The age-related decline of the residents’ immunity resulted in 65 of the 68 residents (95.58%) being infected with symptoms, whereas 34.9% required hospitalization, and 25.8% died. The best-fitting model to explain the distribution of cases reflects three points at the time of infection.”

3 Surveillance of COVID-19 in LTCF in the EU/EEA and the US have reported out- 53 breaks in more than half of these centers [2–4], with incidences from 22% to 47% in resi- 54dents and between 24% and 45% in staff. The CFR ranged from 27% to 36.9% in residents 55[5]. In Spain, the rate per 10,000 residents confirmed with COVID-19 by diagnostic tests 56 over the total number of residents has ranged from 190.92 in January 2022 to 18.6 in Janu- 57ary 2023, while the CFR was 7.65 % and ranged between 20.39% in 2020 and 2.59% in (Please add updated information)

Thanks for the comment. The report about weekly information on COVID-19 in residential centers, which included information on residential centers for the elderly, ceased to be published in February 2023. Since this report was published for the duration of the pandemic situation, there is no current information about these data. However, to clarify this issue, we have added this statement in the introduction section:

In Spain, the weekly report on COVID-19 in residential centers ceased to be published in February 2023. This report displayed that the rate per 10,000 residents confirmed with COVID-19 by diagnostic tests over the total number of residents has ranged from 190.92 in January 2022 to 18.6 in January 2023, while the CFR was 7.65 % and ranged between 20.39% in 2020 and 2.59% in 2023”.

4 PLease modify Study design and setting description 82

We conducted a retrospective observational study on an elderly care facility in León, 83 Spain. We investigated the epidemiological and clinical characteristics of the cohort of a 84COVID-19 outbreak between 30 December 2020 and 15 February 2021 (14 days after the 85 last case). 86The facility consists of one building, with a total capacity of 74 residents living in 87single or double rooms. The useful square meters of the residence were 3785. The occu- 88pancy percentage was 90% (70 of 74) during the outbreak. The staff included 21 geriatri- 89cians, one full-time nurse and occupational therapist, one medical doctor (general practi- 90tioner), one physiotherapist, one social worker and one part-time hairdresser, the Direc- 91tor-Manager, three kitchen staff, three cleaners, and one maintenance worker. The strict 92anti-COVID protocol in the facility included the exclusive lounge utilization for depend- 93ent residents and a second lounge converted into a dining room for the dependents. (ITS NOT CLEAR)

We acknowledge the suggestion. This paragraph has been modified as follows:

We conducted a retrospective observational study on an elderly care facility in León, Spain. We investigated the epidemiological and clinical characteristics of the cohort of a COVID-19 outbreak between 30 December 2020 and 15 February 2021 (14 days after the last case).

The facility consists of one building, with a total capacity of 74 residents living in single or double rooms. The useful square meters of the residence had a surface of 3785 m2. The occupancy percentage was 92% (68 of 74) during the outbreak. The residence had a surface of 3785 m2. The occupancy percentage was 92% (68 of 74) during the outbreak. The number of workers at that moment was 35. The staff composition according to the category was: 25 health workers (21 geriatricians, one full-time nurse and occupational therapist, one medical doctor/general practitioner), one physiotherapist, and one social worker); the director-manager, one part-time hairdresser, three kitchen staff, three cleaners, and one maintenance worker.

7 A chronological account of the events related to this outbreak, including the epidemic 117 curve, was made. The estimation of cases distribution was according to the incubation 118 periods reported by McAloon et al. [18], calculating the proportions of hospitalization and 119 death with their 95% CI and their distribution by age, sex, estimated date of infection, time 120 from vaccination to onset of symptoms, and Cycle threshold of RT-PCR. Using uncondi-121 tional logistic regression, we estimated the risks of hospitalization and death adjusted for 122 sex, age, time from vaccination to symptom onset, and PCR threshold cycles. 123

The above-mentioned text has been revised.

2.4. Ethical consideration 124

The clinical research ethics committee of the Hospital of León approved the study 125 protocol. Informed consent was not required. The information safety commission pro-126 vided approval. The study followed the principles of the Declaration of Helsinki.(A chronological account of the events related to this outbreak, including the epidemic 117 curve, was made. The estimation of cases distribution was according to the incubation 118 periods reported by McAloon et al. [18], calculating the proportions of hospitalization and 119 death with their 95% CI and their distribution by age, sex, estimated date of infection, time 120 from vaccination to onset of symptoms, and Cycle threshold of RT-PCR. Using uncondi-121 tional logistic regression, we estimated the risks of hospitalization and death adjusted for 122 sex, age, time from vaccination to symptom onset, and PCR threshold cycles. 123

2.4. Ethical consideration 124The clinical research ethics committee of the Hospital of León approved the study 125 protocol. Informed consent was not required. The information safety commission pro-126 vided approval. The study followed the principles of the Declaration of Helsinki.(PLEASE PROVIDEethical clearance AUTHORISED  CERTIFICATE AND UPDATE WITH NEW INFORMATION)

Thanks for the comment. We have included this information in the revised version of the manuscript:

“The research protocol of this study was approved by the Ethical Committee for Research with Medicines of the León and Bierzo Health Areas (CEIm) (P.I. 20122).The clinical research ethics committee of the Hospital of León approved the study protocol. Informed consent was not required. The information safety commission provided approval. The study followed the principles of the Declaration of Helsinki”.

9On the 9th, three more residents presented positive results in 136 the antigen detection tests. On this day, samples for RT-PCR taken from the residents of 137 the center and the workers resulted in 51 of the 68 residents (75.0%) being found positive 138 while 18 of the 32 workers (56.3%) tested positive. At that time, the residence was sector-139 ized, leaving one floor for positive cases and another for negative ones. 140

The 17 negative cases among the residents were re-sampled after seven days, and 14 141 tested positive. Altogether 65of the 68 residents (95.58%) were infected with SARS-CoV-142 2. All 65residents with PCR+ had symptoms. Figure 1 shows the distribution of cases by 143 date of symptom onset.(PLEASE MODIFY )

FIGURE 1 AND FIGURE 2 PLEASE CHANGE COMPLETELY (please add hours and logic of taking 7,11 please explain)

Figures 1 and 2 have been revised and modified to improve the presentation of results.

11,Data availability statement is missing

This information is included.

12 Pleaee arrange the references clearly as per journal format and check with mendeley 0r zotero

The references have been conveniently revised and modified in the revised version.

Best

Thank you so much for the positive comments.

Reviewer 2 Report

This was an interesting report on an outbreak of Covid-19 that affected a very large percentage of residents in an elderly care facility an sadly resulted in many fatalities. While the data presented were well described and the conclusions reached were sound, to my thinking this case report only provides part of the story.  In the Introduction you describe how ECDC and CDC guidelines recommended priority vaccination but also to maintain control measures after starting vaccination, but how does that translate to specific legislation or guidance in Spain? Were there instructions with this brand of vaccine about the delay time before efficacy? Was there any communication from the Spanish government to care facilities about vaccination and the need to maintain control measures? And if so, how was that applied (or not) at this facility? No details of control measures applied in this facility are described other than the segregation protocol lines 92-97. It would be useful to know more about other hygiene controls in place up until vaccination was started and whether these were changed once vaccination started. Presumably (it isn't stated) there were no cases of Covid-19 in residents or staff before this outbreak? In which case something happened that coincided with the start of the vaccination programme to introduce the disease into this community, but the only evidence put forward (lines 208 - 210) is that it may have been to do with Christmas holidays and returns from weekends but that is not supported by any data. Would it be possible to access anonymous records of staff (presumably not residents) movements in and out of the facility? This may support the hypothesis and would therefore give greater impact to the paper in terms of the message that it is important for residential facilities for the elderly to maintain strict hygiene controls and that vaccination alone is not a sufficient control measure. In the hierarchy of control, vaccination is at the bottom, below (in order) elimination, engineered controls, procedural controls and PPE.               

English language fine - minor edits needed.

Author Response

We thank the reviewers for their constructive and valuable suggestions. Their careful reading has contributed to improving the quality of this article. We have comprehensively revised the manuscript to address all their requests. Following the editor’s instructions, we have included a point-by-point response to the reviewers’ comments, and the changes made have been highlighted accordingly in the revised version of the manuscript.

To help review amendments done to the manuscript, below we reproduce the reviewer’s comments (black color and italics font), and immediately after, the reply is provided in red color and portions of the revised text in the paper between quotation marks (red color and italics).

Reviewer 2

This was an interesting report on an outbreak of Covid-19 that affected a very large percentage of residents in an elderly care facility an sadly resulted in many fatalities. While the data presented were well described and the conclusions reached were sound, to my thinking this case report only provides part of the story.

In the Introduction you describe how ECDC and CDC guidelines recommended priority vaccination but also to maintain control measures after starting vaccination, but how does that translate to specific legislation or guidance in Spain?

We thank the reviewer for the careful reading and positive comments. In response to this suggestion, we have included additional information about the COVID-19 Vaccination strategy in Spain.

“Following the European Union vaccine strategy to ensure rapid and equitable access to vaccines by all Member States, the COVID-19 Vaccination Technical Working Group and the Vaccine Committee coordinated by the Inter-territorial Council of the National Health System (CISNS) adopted in Spain a set of measures to prioritize the vaccination of population groups according to an established ethical framework and risk criteria. Based on the risk of severe morbidity and mortality, exposure, socioeconomic impact, and transmission, the group prioritization established that the first stage of vaccination might include residents and health and social care personnel in care homes for the elderly and the disabled”.

Were there instructions with this brand of vaccine about the delay time before efficacy?

The reviewer´s suggestion is appreciated. We have completed the information about the vaccine (Comirnaty®, BioNTech/Pfizer) by including the authorization and product information contained in the European Medicines Agency after approving the administration of this medicine in the Introduction:

“The European Medicines Agency has authorized the use of the mRNA according to their vaccine efficacy information”.

 Additional information about the delay time before efficacy is described in the discussion section (lines 253-255): Although clinical trials of the BNT162b2 vaccine have reported an efficacy of 52% after administration of the first dose [28,29], these efficacies appeared between 13 to 24 days after immunization of vaccination [29].

Was there any communication from the Spanish government to care facilities about vaccination and the need to maintain control measures? And if so, how was that applied (or not) at this facility?

In Spain, the Territorial Council of Social Services and the System for Autonomy and Care for Dependency publish weekly a statistic on the situation of residential centers. These data are part of the protocol for epidemiological surveillance of residential centers in the EU/EEA countries coordinated by the European Centre for Disease Prevention and Control (ECDC) whose metadata began to be implemented at the end of January 2021 in the TESSy system (The European Surveillance System). The collection and systematization work is carried out jointly by the Instituto de Mayores y Servicios Sociales (IMSERSO) of the Ministry of Social Rights and Agenda 2030, Centro de Coordinación de Alertas y Emergencias Sanitarias (CCAES), of the Ministry of Health) and the Instituto de Salud Carlos III of the Ministry of Science and Innovation. They are being regularly submitted for subsequent publication at: https://www.ecdc.europa.eu/en/Covid-19.

The care facility object of this study fulfilled the recommendations on vaccination and control measures. To clarify this aspect, we have included the following statement in the method section:

“The care facility followed the recommendations on vaccination and control measures contained in the protocol for epidemiological surveillance of residential centers in the EU/EEA countries coordinated by the European Centre for Disease Prevention and Control (ECDC) [3] as well as the recommendations to nursing homes and social-health centers for COVID-19 of the Spanish government [18].”

No details of control measures applied in this facility are described other than the segregation protocol lines 92-97. It would be useful to know more about other hygiene controls in place up until vaccination was started and whether these were changed once vaccination started. Presumably (it isn't stated) there were no cases of Covid-19 in residents or staff before this outbreak?

We thank the reviewer`s suggestion. The following statement has been added to complete this information:

“These measures included not only actions to be taken in case of contacts and cases of COVID-19 but also general measures for the protection of workers' health including hand hygiene, use of personal protective equipment (e.g., gloves, masks, eyewear), cleaning and disinfection of surfaces and spaces, waste management, crockery and linen, and identification of contacts of cases under investigation”.

In which case something happened that coincided with the start of the vaccination programme to introduce the disease into this community, but the only evidence put forward (lines 208 - 210) is that it may have been to do with Christmas holidays and returns from weekends but that is not supported by any data. Would it be possible to access anonymous records of staff (presumably not residents) movements in and out of the facility? This may support the hypothesis and would therefore give greater impact to the paper in terms of the message that it is important for residential facilities for the elderly to maintain strict hygiene controls and that vaccination alone is not a sufficient control measure. In the hierarchy of control, vaccination is at the bottom, below (in order) elimination, engineered controls, procedural controls and PPE.

We strongly agree with the reviewer. The movements of the staff in and out of the facility according to their work shifts might contribute to the spread of transmission. Although the worker’s schedule register was available, the surveillance system could not find a transmission trend between staff and residents, and molecular epidemiology was not available at these moments. This fact is explained in the text:

“The high prevalence of infection in residents and workers makes it difficult to know whether transmission occurred from workers to residents or vice versa. The staff might contribute to the rapid circulation of the virus due to non-ordinary circumstances, such as the family gatherings surrounding the time of infection. Additionally, the maintenance at work of staff with early minor symptoms could result in delayed identification. Recent vaccination and the possible adverse effects, the atypical clinical presentation in those vaccinated, could also have affected the delay in taking measures and the late identification of the index case(10.5 days after vaccination) [24,25]”.

 However, to emphasize this point of view, the revised version of the manuscript contains the reviewer’s suggestion about the hierarchy of control measures in long-term care facilities involving from most to least important: elimination, engineered controls, procedural controls, PPE, and vaccination, as reported in the Recommendations to nursing homes and social-health centers for COVID-19 of the Ministry of Health of the Spanish government. The new statement is included in the discussion section as follows:

“This fact revealed the importance of maintaining strict hygiene controls regardless of initial vaccination since engineered and procedure controls and the appropriate use of personal protective equipment (PPE) outweigh the role of vaccination for the infection prevention and control of COVID-19 in elderly nursing homes”.

Round 2

Reviewer 2 Report

Thank you for the revisions included in this latest draft and for explaining them. In my view this has now addressed my previous comments and suggestions.

Very minor - repetition of words in redrafted sentence at line 56. 

Author Response

Thanks for your comment.

The repetition of words in a redrafted sentence at line 56 has been deleted.